# Seasonal Variations in the Lipid Profile of the Ovarian Follicle in Italian Mediterranean Buffaloes

**DOI:** 10.3390/ani12162108

**Published:** 2022-08-17

**Authors:** Michal Andrzej Kosior, Alfonso Calabria, Maria Paz Benitez Mora, Marco Russo, Giorgio Antonio Presicce, Natascia Cocchia, Salvatore Monti, Hilde Aardema, Bianca Gasparrini

**Affiliations:** 1Department of Veterinary Medicine and Animal Production, Federico II University, Via F. Delpino 1, 80137 Naples, Italy; 2Faculty of Veterinary Sciences, National University of Asunción (U.N.A.), San Lorenzo 2169, Paraguay; 3ARSIAL, Centro Sperimentale per la Zootecnia, Via Lanciani 38, 00162 Rome, Italy; 4Department of Veterinary Sciences, University of Messina, 98168 Messina, Italy; 5Department of Farm Animal Health, Faculty of Veterinary Medicine, Utrecht University, Yalelaan 7, 3584 CL Utrecht, The Netherlands

**Keywords:** buffalo, lipid, 1H-NMR, competence, oocyte

## Abstract

**Simple Summary:**

Reproductive seasonality is a major factor affecting buffalo breeding. The rationale of this work derives from the hypothesis that the reduced cleavage and blastocyst rates observed during the non-breeding season could be due to a suboptimal follicular environment. The present study aimed to evaluate the influence of season on the lipid profile of the ovarian follicle in the Italian Mediterranean buffalo. For this purpose, abattoir-derived ovaries were collected during the breeding and non-breeding seasons, and the apolar phase of follicular components was analyzed. To our knowledge, this is the first report of seasonal variations in lipid content of the buffalo ovarian follicle, including follicular fluid, follicular and cumulus cells, and oocytes. The results undoubtedly demonstrated significant seasonal variations in the lipid profile, including triglycerides, cholesterol, and phospholipids, in the different biological matrices analyzed. Interestingly, an increased amount in the total level of non-esterified fatty acids in the follicular fluid was also observed during the non-breeding season. The results allow a better understanding of the physiology of the ovarian follicle in buffalo and unveil some causes of reduced oocyte competence during the non-reproductive season, laying the groundwork for further studies and corrective strategies.

**Abstract:**

The reduced oocyte competence recorded during the non-breading season (NBS) is one of the key factors affecting the profitability of buffalo farming and limits the IVEP efficiency. The purpose of this experiment was to evaluate whether season influences the lipid content within the ovarian follicle in the Italian Mediterranean buffalo. Abattoir-derived ovaries were collected during the breeding season (BS) and the NBS, and different matrices (follicular fluid, oocytes, cumulus and follicular cells) were recovered. After the extraction of the apolar fraction, all samples were analyzed by H1 nuclear magnetic resonance and FF samples by gas chromatography–mass spectrometry. Seasonal differences in lipid composition were observed in all matrices. In particular, during the NBS, the triglyceride content was higher in the follicular fluid and in the oocytes but reduced in the follicular cells. Both cholesterol and phospholipids were reduced in the follicular fluid and follicular cells during the NBS. Furthermore, the total amount of non-esterified fatty acids was significantly increased in the follicular fluid. The seasonal variation in lipid profile of the follicle may, in part, account for the reduced buffalo oocyte competence during the NBS, due to the critical role played by lipids in regulating ovarian functions.

## 1. Introduction

The importance of buffalo (Bubalus bubalis) breeding in the world is proven by the constant increase in the number of buffalo heads reared worldwide [1]. This trend is due to the fundamental role played by the species as an important livestock resource, especially for tropical and subtropical countries.

Reproductive seasonality is one of the key factors limiting the profitability of intensive farming of this species, resulting in the discontinuity of milk market supply and limiting the utilization of reproductive technologies, which are currently essential tools for genetic improvement. In Italy, where buffalo breeding is essentially tied to the production of mozzarella cheese, the pattern of seasonality, with increased reproductive activity during months characterized by decreasing day length, is opposite to the market demand. Therefore, in order to modify the calving calendar, the out-of-breeding mating strategy (OBMS) is often applied [2]. This, however, results in reduced fertility and a higher proportion of embryonic mortality [3], partly attributable to impaired luteal function [4] and partly to reduced oocyte competence, as shown by decreased blastocyst yields after in vitro fertilization [5]. In addition, lower intrafollicular levels of estradiol and IGF-1 were found in association with a decline of oocyte quality during the non-breeding season (NBS) in Murrah buffalo heifers [6]. Furthermore, it has recently been reported that season influences the miRNAs and transcriptomic profile of oocytes and follicular cells in buffaloes [7]. Interestingly, the gene ontology analysis revealed a seasonal variation of genes involved in pathways related to triglyceride and sterol biosynthesis and storage in the oocyte, implying a possibly altered lipid profile of the ovarian follicle. These findings stress the importance to evaluate the lipid profile of the follicle in relation to season with the aim to unveil the causes of the reduced oocyte competence recorded in buffalo during the NBS.

Lipids are hydrophobic organic molecules with an essential role in cell metabolism. Lipids are a primary energy source for cells, are involved in cell signaling, and are essential constituents of cell membranes [8,9]. Moreover, lipids are also precursors for synthesis of steroids and eicosanoids in the ovarian follicle [10]. The lipid composition of the various components of the ovarian follicle is influenced by countless factors such as age, diet, state of nutrition, metabolic status, and environmental temperature, but also varies according to the stage of follicular development [6,11,12]. The lipid composition of the follicular fluid and the cellular component of the follicle is expected to be influenced by the plasma composition, as has been demonstrated for bovines [12,13,14], but also by the metabolism of the follicular cells. During follicle growth, lipid synthesis, transport, storage, and degradation occur in the oocyte and surrounding cells [15,16]. It is known that the granulosa and cumulus cells support the oocyte during development; in particular, cumulus cells are able to protect the oocyte against elevated levels of non-esterified fatty acids (NEFA) (mainly palmitic and stearic) in bovines [17]. Furthermore, the cumulus cells appear to modulate the transfer of fatty acids towards the oocyte [18,19].

Therefore, the purpose of this experiment was to evaluate whether the season influences the lipid content within the ovarian follicle in the Italian Mediterranean buffalo, to understand the causes of the reduced competence during the NBS. In order to achieve the objective, the lipid profile of the buffalo ovarian follicle, specifically follicular fluid (FF), follicular cells (FC), cumulus cells (CC), and oocytes (OO), was analyzed in relation to season by 1H nuclear magnetic resonance (NMR) spectroscopy. The total NEFA levels were assessed by gas chromatography–mass spectrometry (GC–MS).

## 2. Materials and Methods

### 2.1. Experimental Design

The samples were collected in October, i.e., during the breeding season (BS) and in January, i.e., during the NBS, over a total of 8 replicates (4/season). The ovaries were collected at a local slaughterhouse, under national food hygiene regulations from cyclic multiparous Italian Mediterranean buffalo cows (on average 40 per season), grown under controlled nutrition and housed inside barns in intensive farms located in the province of Caserta, Italy (latitude 40.5–41.5° and longitude 13.5–15.5°). The age and weight of the animals were 4.8 ± 0.6 years and 547.2 ± 14.3 kg, respectively. Cyclic ovarian activity was assessed by two clinical examinations carried out 12 days apart before slaughter, to detect the presence of a follicle greater than 1 cm and/or corpus luteum on the ovary. The ovaries of healthy and cycling animals were transported to the laboratory within 4 h after slaughter in physiological saline supplemented with 150 mg/L kanamycin at 30–35 °C. During each season, four replicates were carried out to collect FF, FC, CC, and OO samples. Briefly, on the day of slaughter, follicular fluid was aspirated, separated from follicular cells, and half of the COCs found were denuded in order to collect both oocytes and cumulus cells; then, these samples were stored at −80 °C until analyses. The remaining COCs were in vitro matured, fertilized, and cultured according to our standard protocol [11,20], briefly described below, to assess developmental competence.

### 2.2. Collection of Follicular Fluid and Follicular Cells

In the laboratory, the ovaries were washed three times with physiological saline with 150 mg/L kanamycin to remove blood residues. Follicular fluid was aspirated from 3–5 mm diameter follicles using an 18 G needle under vacuum (40–50 mm Hg) and collected in Falcon tubes. Follicular fluid from a pool of 20 follicles was poured into a Petri dish for oocyte recovery. All the COCs found were allocated to a dish with HEPES-buffered TCM199 supplemented with 10% fetal calf serum (FCS; H199) and evaluated according to morphology and classified according to Di Francesco et al. (2011) [20]. The remaining fluid was transferred into 1.5 mL Eppendorf tubes and centrifuged (300× *g* for 10 min). The supernatant was then stored as follicular fluid (FF). The remaining pellet was washed in PBS and subjected to two consecutive centrifugations (2000× *g* 10 min), then the supernatant was removed and the pellet stored at −80 °C as follicular cells (FC). Grade A and B COCs, considered suitable for in vitro embryo production (IVEP), were quickly selected from the dish and washed thoroughly in medium H199, while the other oocyte categories were discarded.

### 2.3. Collection of Cumulus Cells and Oocytes

For each replicate, A and B COCs were split into two groups, as previously described. In the first group, the oocytes were pooled into groups of 10, then moved into vial tubes in PBS and vortexed for 3 min. Then, the content of the tubes was moved to a Petri dish; the oocytes were removed and transferred into Eppendorf tubes in minimum volume and stored at −80 °C as oocytes (OO). The remaining liquid was centrifuged at 2500× *g* 15 min and then, after removing the supernatant, the pellet was stored at −80 °C as cumulus cells (CC).

### 2.4. In Vitro Embryo Production

For each replicate, Grade A and B COCs recovered by follicular aspiration were rinsed in H199 medium and in vitro matured. The methods for in vitro maturation (IVM) described below have been reproduced, in part, from Gasparrini et al. (2000) [21]. Briefly, COCs were allocated to 50 µL drops (10 per drop) of IVM medium, i.e., in TCM199 buffered with 25 mM sodium bicarbonate and supplemented with 10% FCS, 0.2 mM sodium pyruvate, 0.5 µg/mL FSH, 5 µg/mL LH, 1 µg/mL 17 β-estradiol, and 50 µg/mL kanamycin and incubated at 38.5 °C for 21 h in a controlled gas atmosphere of 5% CO2 in humidified air. The methods for in vitro fertilization (IVF) and culture (IVC) described below have been reproduced from Di Francesco et al. (2012) [5]. Frozen straws from a bull previously tested for IVF were thawed at 37 °C for 40 s and sperm was selected by centrifugation (25 min at 300× *g*) on a Percoll discontinuous gradient (45% and 80%). The sperm pellet was re-suspended to a final concentration of 2 × 10^6^ mL^−1^ in the IVF medium, consisting of Tyrode albumin lactate pyruvate supplemented with 0.2 mM penicillamine, 0.1 mM hypotaurine, and 0.01 mM heparin. Insemination was performed in 50 µL drops of IVF medium under mineral oil (5 oocytes per drop) at 38.5 °C under humidified air with 5% CO_2_. Twenty hours after IVF, presumed zygotes were removed from the IVF medium, stripped of cumulus cells by gentle pipetting, and allocated to 20 µL drops under mineral oil of IVC medium, i.e., synthetic oviduct fluid (SOF) including essential and non-essential amino acids and 8 mg/mL bovine serum albumin [22]. Culture was carried out under humidified air with 5% CO_2_, 7% O_2_, and 88% N_2_ at 38.5 °C. On day 5 post-insemination (pi), the cleavage rate based on the number of oocytes was assessed, and the embryos were transferred into fresh medium until day 7 of IVC (end of culture). The embryos were scored for quality on the basis of morphological criteria [23], and the percentages of transferable embryos (Grade 1 and 2 tight morulae and blastocysts) and of Grade 1 and 2 blastocysts were recorded.

### 2.5. Extraction of the Apolar Fraction from Samples for 1H-NMR

The separation of the polar and the apolar fractions of each type of sample was carried out by a method previously described (Santonastaso et al., 2017) [24]. Briefly samples were first re-suspended in 170 µL of H_2_O and 700 µL of methanol. For cell samples (FC, OO and CC) the solution was sonicated for 30 s to lyse the membranes, cell lysis was confirmed under optical microscope and lysates were further processed, while sonication was not carried out for cell-free samples (FF). Then, 350 µL of chloroform was added and the samples were mixed on an orbital shaker in ice for 10 min. After this, 350 µL of a 1:1 (*v/v*) H_2_O/chloroform solution was added to each sample, the samples were vortexed for 5 sec and centrifuged at 4000 rpm for 10 min at 4 °C. Following centrifugation three different phases were separated: an upper phase (containing polar metabolites), a middle phase (with cell debris, denatured proteins and RNA) and a lower phase (containing apolar metabolites). Therefore, the lipophilic (apolar) phase was collected, evaporated and stored at −80° until analysis.

### 2.6. 1H-NMR Metabolomic Analysis

The methods for 1H-NMR metabolomic analysis have been reproduced from Santonastaso et al. (2017) [24]. The apolar fractions were dissolved in 630 µL of PBS-D_2_O with the pH adjusted to 7.2, and 70 µL of sodium salt of 3-(trimethylsilyl)-1-propanesulfonic acid (1% in D2O) was used as the internal standard. A 600 MHz Avance Bruker spectrometer with a TCI probe was used to acquire 1H-NMR spectra at 300 K. An excitation sculpting pulse sequence was applied to suppress the water resonance. A double-pulsed field gradient echo was used, with a soft square pulse of 4 ms at the water resonance frequency and gradient pulses of 1 ms each in duration adding 128 transients of 64k complex points, with an acquisition time of 4 s per transient. Time-domain data were all zero-filled to 256k complex points and an exponential amplification of 0.6 Hz prior to Fourier transformation was applied.

All of the 1H-NMR spectra were automatically phased using the “apk” command in Topspin 4.0 (Bruker, Biospin, Germany), which performs an automatic phase correction using both a zero- and first-order correction. In a few cases, we manually optimized the spectra phasing. Then, we performed a baseline correction using the “absn” command in Topspin 4.0 (Bruker, Biospin, Germany), which automatically fits the spectra baseline to a polynomial of degree given by the processing parameter absg (usually 5). We referenced the spectra to the CH3 resonance of TSP at 0 ppm. The spectral 0.50–8.60 ppm region of 1H-NMR spectra was integrated in buckets of 0.04 ppm by the AMIX package (Bruker, Biospin, Germany). We normalized, via the MetaboAnalyst v5.0 tool [25], the bucketed region using a normalization procedure grouped into two categories: (i) sample normalization, that is, for general-purpose adjustment for systematic differences among samples; and (ii) data scaling which adjusts each variable/feature by a scaling factor computed based on the dispersion of the variable. Specifically, we normalized by sum (the total spectrum area) and used Pareto scaling (mean-centered and divided by the square root of the standard deviation of each variable).

### 2.7. Extraction of Apolar Samples for GC/MS

The samples were extracted by using a modification of a protocol previously described [26]. Briefly, samples were extracted by using chloroform and 0.1% H_2_SO_4_, incubated with shaking for 1 h, and then centrifugated at 10,000 rpm for 15 min. The analytes soluble in chloroform were dried under a nitrogen stream. The NEFAs were extracted using 99/1 diethyl ether/glacial acetic acid (*v/v*) and the solvent was evaporated under vacuum. The samples were mixed with 100 μL *N*,*O*-Bis(trimethylsilyl)trifluoroacetamide (BSTFA). The reaction was conducted for 30 min at 90 °C. Finally, the samples were dried, reconstituted in 50 μL of hexane, and analyzed by GC/MS.

### 2.8. GC/MS Analysis

GC/MS analysis was performed by a 7820A (Agilent Technologies, Santa Clara, CA, USA) with a HB-5ms capillary column (30 m × 0.25 mm × 0.25 µm film thickness) (Agilent Technologies). The injector, ion source, quadrupole, and the GC/MS interface temperature were 230, 230, 150, and 280 °C, respectively. The flow rate of helium carrier gas was kept at 1 mL/min. An amount of 1 µL of derivatized sample was injected with a 3 min solvent delay time and split ratio of 10:1. The initial column temperature was 90 °C, held for 2 min, ramped to 150 °C at the rate of 15 °C/min and held 1 min, and then finally increased to 280 °C at the rate of 30 °C/min and kept at this temperature for 5 min. The ionization was carried out in the electron impact (EI) mode at 70 eV.

The MS data were acquired in full-scan mode from *m/z* 40–400 with acquisition frequency of 12.8 scans per second. The identification of compounds was confirmed by injection of pure standards and comparison of the retention time and corresponding EI MS spectra. The contents of fatty acids were calculated with external standard methods.

### 2.9. Statistical Analysis

Orthogonal projections to latent structure-discriminant analysis (OPLS-DA) and S-Plot were used by the Metabo Analyst tool to analyze differences between seasons in different types of samples because it can effectively deal with chemical shift variation in full-resolution 1H-NMR datasets without the need of binning or alignment steps [27]. Furthermore, the variable importance in projection (VIP) method was used to identify specific metabolites that show the greatest variations between seasons and their patterns.

The metabolites that were selected had a VIP score higher than 1. The predictive ability of the model was estimated by using Q2, which was calculated via cross-validation (CV). The identification and relative quantification of the metabolites was carried out by the Chenomx Profiler [28].

Differences in total NEFA, palmitic, and stearic acids were analyzed by a Student’s t-test, while the differences in cleavage and blastocyst percentages between seasons were analyzed by a chi-square test, setting the level of significance at *p* < 0.05.

## 3. Results

### 3.1. In Vitro Embryo Production

Oocyte developmental competence was significantly influenced by the season. During the NBS, the cleavage rate was decreased compared to the BS (60.7% vs. 76.8%; *p* < 0.01). Furthermore, during the NBS, the percentage of total transferable embryos and of Grade 1 and 2 blastocysts were decreased (17.9% and 16.2%, respectively; *p* < 0.05), in comparison to the BS (28.6%; *p* < 0.05 and 26.2%, respectively; *p* < 0.05).

### 3.2. 1H-NMR Metabolomic Analysis of Follicular Fluid

The OPLS-DA plot showed that the NMR spectra of follicular fluid (FF) samples collected during the BS and the NBS clustered separately, indicating significant differences in proton signals (metabolites) between the seasons (Figure 1A). The VIP plot revealed the 15 metabolites with the highest seasonal variation in FF (Figure 2A). In particular, during the NBS, the FF content of triglycerides was higher, while that of cholesterol and phospholipids was lower and seasonal differences were observed in the profile of some FAs.

### 3.3. 1H-NMR Metabolomic Analysis of Follicular Cells

The follicular cells (FC) samples also showed a clear separation of the NMR spectra, as shown in Figure 1B, suggesting seasonal differences in the lipid profile, highlighted in the VIP plot that shows the 15 metabolites with a VIP score of more than 1 (Figure 2B). In the FCs, the majority of lipids (cholesterol, triglycerides, phospholipids, and some FAs) were reduced during the NBS, with the exception of an unidentified FA.

### 3.4. 1H-NMR Metabolomic Analysis of Cumulus Cells

With regard to the cumulus cells (CC), the OPLS-DA plot showed that the samples collected in different seasons clustered separately, even though NMR spectra slightly overlapped (Figure 1C). As shown in the VIP plot (Figure 2C), higher levels of cholesterol and arachidonic acid, and lower levels of omega-3 FA and phospholipids were found during the NBS (Figure 2C).

### 3.5. 1H-NMR Metabolomic Analysis of Oocytes

Finally, differences in lipid profile were found in oocytes (OO), as shown by the clusterization reported in Figure 1D and by the VIP plot (Figure 2D). More specifically, the amounts of phospholipids, glycerophospholipids, and triglycerides (TAG) were increased during the NBS, while levels of cholesterol and of some FAs were reduced.

In Table 1, the identified lipids corresponding to proton signals showing differences between seasons in the different components of the ovarian follicle are summarized.

### 3.6. Analysis of Non-Esterified Fatty Acids in the Follicular Fluid by GC–MS

In the NBS, the content of total NEFA in FF was increased compared to the BS (288.8 ± 84.1 vs. 31.4 ± 7.0 mg/L; *p* < 0.01). The most abundant NEFAs in the buffalo FF, i.e., saturated palmitic and stearic acids, showed the same seasonal trend. The levels in the NBS were, for palmitic, 195.6 ± 9.8 mg/L and, for stearic acid, 86.0 ± 30.0 mg/L acid versus, respectively, 11.1 ± 2.0 mg/L and 3.9 ± 0.4 mg/L (*p* < 0.01) in the BS.

## 4. Discussion

The present study aimed to evaluate whether season influences the lipid profile of the ovarian follicle in the Italian Mediterranean buffalo, starting from the hypothesis that the reduced oocyte competence recorded during the NBS may originate from a suboptimal follicular environment. To our knowledge, this is the first report of seasonal variations in lipid content in the different components of the follicle, including the follicular fluid, follicular and cumulus cells, and the oocyte.

During the NBS, reduced cleavage and blastocyst yields were observed after IVF of buffalo oocytes, confirming a previously demonstrated seasonal influence on oocyte competence [5,7,20]. In addition to the seasonal trend, the blastocyst rates obtained in this work are comparable to those reported in the literature from abattoir-derived oocytes in the Mediterranean buffalo [5,7,20]. The results of the current work demonstrate significant seasonal variations in the lipid profile of the different biological matrices that were analyzed. In addition, as the MNR analysis showed only a relative quantification and did not allow us to identify individual NEFAs, GC–MS was carried out on FF to assess the NEFA fraction.

Interestingly, in the FF, a significantly higher level of NEFAs, and the individual NEFAs palmitic and stearic acids, were observed in the NBS in comparison to the BS. The increased level of NEFAs in FF suggests increased mobilization of body fat reserves in buffaloes during the NBS. The elevated level of NEFAs matches with the formerly observed elevated level of the metabolite beta-hydroxybutyrate (BHBA) in the FF during the NBS (data not shown). This indicates a condition of energy scarcity and suggests a negative energy balance (NEB) status in the buffaloes during the NBS. It is worth highlighting that the NBS in Italy corresponds to the months characterized by low environmental temperatures, to which buffaloes are highly sensitive due to their tropical origin; hence, they may undergo NEB as a result of impaired thermoregulation.

Elevated levels of NEFAs, in particular, saturated NEFAs, can have a negative impact on oocyte developmental competence in dairy cows [12,22]. Former studies demonstrated an accumulation of triglyceride (TAG) in the lipid droplets of various somatic cells exposed to elevated levels of NEFAs for storage of potentially toxic fatty acids, a phenomenon which was stimulated by the presence of unsaturated oleic acid [23,24,27]. Furthermore, a massive accumulation of lipids in granulosa and cumulus cells has been demonstrated in response to elevated NEFA levels in follicular fluid [13,27]. The current study showed an increase of TAG in oocytes, probably a consequence of an exposure to the elevated level of NEFAs and TAG during the NBS. It is known that the lipid composition of the extracellular medium can influence the lipid composition of oocytes [29]. Bovine oocytes that are matured in vitro with serum, rich in lipids, show a higher intracytoplasmic content of TAG and cholesterol compared to those matured in serum-free media [30]. The elevated levels of both NEFAs and TAG found in the FF in the current study may have contributed to the increased level of TAG observed in the oocyte during the NBS. The fatty acids that are stored as TAG in lipid droplets can be used as an endogenous energy source and are important for the formation of phospholipid membranes by the oocyte and embryo in different species, including the buffalo [31,32]. However, an excessive accumulation of lipids has been related with a negative effect on oocyte competence and cryotolerance [8].

The uptake of fatty acids by bovine oocytes from medium has been demonstrated before and appears to be strongly regulated by the surrounding cumulus cell layer [18,19,22]. Bovine cumulus cells are able to protect the oocyte against a high level of NEFAs, by massive storage of fatty acids in lipid droplets, while the lipid content in oocytes is unaffected [13,18]. In the current study, there was no indication for an accumulation of TAG in cumulus cells of the buffalo, as there was no increase in the TAG amount in these cells during the NBS. This could suggest a reduced potential of cumulus cells to store fatty acids in the presence of elevated levels of NEFAs in the buffalo to protect the oocyte, and may explain the elevated levels of TAG detected in the buffalo oocyte in the current study. This may be accounted for by species-specific differences, as the buffalo oocyte is surrounded by fewer layers of cumulus cells that show lower adhesion [33].

An intriguing finding of the current study was the higher level of saturated palmitic and stearic acids found in FF by GC–MS analysis during the NBS. In fact, saturated NEFAs are able to induce lipotoxic events in several somatic cell types, including granulosa and cumulus cells [12,17,18,23,34,35]. The type of free fatty acid, saturated or unsaturated, to which oocytes are exposed also appears to have a major impact on the developmental competence of the bovine oocyte. Saturated palmitic and stearic acids have a dose-dependent negative impact on the potential of the bovine oocyte to develop into a blastocyst [12,22]. The negative impact of the potentially toxic saturated palmitic and stearic acids is counteracted by the simultaneous exposure to unsaturated oleic acid [22]. In the current study, no seasonal variations in oleic acid were recorded at the follicular level, but the higher concentrations of both palmitic and stearic acids in the FF may account for the reduced oocyte developmental competence recorded in buffalo during the NBS.

The NMR analysis also showed a difference in the levels of omega-3 fatty acids between the two seasons. During the NBS, the levels of omega-3 fatty acids were lower in FF, FC, CC, and OO. Omega-3 fatty acids have been related to an anti-inflammatory response [36]. Interestingly, an increased level of omega-3 fatty acid in blood has been associated with improved fertility, demonstrated by a higher chance for successful ART in humans [37]. In contrast, the levels of arachidonic acid (AA), an omega-6 polyunsaturated fatty acid, were higher during the NBS in FF and CC. AA is a substrate for prostaglandin synthesis that accounts for approximately 2.5% of the lipid component of bovine follicular fluid [38]. Omega-6 fatty acids are associated with a pro-inflammatory response [38]. The high level of omega-6 and low level of omega-3 fatty acids in the NBS is, thus, certainly worthwhile to investigate more extensively in future studies.

Seasonal differences in cholesterol and phospholipids content were also observed in all analyzed matrices, with lower amounts observed in FF and FC, and for cholesterol a higher level in CC and a lower level in the OO during the NBS, whereas this was the opposite for the level of phospholipids with a low and respectively high level in the CC and OO. Cholesterol and phospholipids are essential elements for the formation of cell membranes and play a crucial role during the rapid cell division process occurring after fertilization. Furthermore, cholesterol plays a role as a substrate for the production of steroid hormones by theca, granulosa, and cumulus cells [39]. Interestingly, anomalies of cholesterol levels in the OO, both in case of excess and defect, can negatively influence maturation, fertilization, and embryonic development. An excess of cholesterol has been associated with infertility in mice due to a premature activation of the oocyte [39]. Sub-physiological levels of cholesterol, on the other hand, in mouse oocytes are associated with a delay in the extrusion of the second polar body and a reduction in fertilization rates [40]. Despite lower concentrations in FF, FC, and CC, phospholipids were higher in the oocyte during the NBS. We may speculate that the increased level of phospholipids indicates a higher level of membranes, possibly due to an increased level of lipid droplets needed for the higher storage of TAG in the oocytes. In any case, an unbalanced cholesterol:phospholipid ratio may lead to altered membrane stability and fluidity in the NBS, with negative consequences on cell survival and, hence, competence [41,42].

## 5. Conclusions

In conclusion, seasonal variations in the lipid profile of various follicle components were demonstrated that may account for the decreased oocyte developmental competence recorded in buffalo during the NBS. In particular, higher triglyceride and NEFA content were found in the FF during the NBS, together with higher amounts of triglycerides in the oocyte, suggesting that lipid mobilization occurs as a result of NEB. Furthermore, it was demonstrated that the most representative FAs detected in the FF, i.e., palmitic and stearic acids, saturated FAs known to affect oocyte competence, were more abundant during the NBS. Furthermore, the observed increased amount of the omega-6 fatty acid arachidonic acid in FF and CC and reduced amount of omega-3 in FF, FC, CC, and OO during the NBS are of interest due to the link with inflammatory responses in the body. Seasonal differences were also observed at different follicular levels in the content of cholesterol and phospholipids, essential for the formation of cell membranes and their stability. These results lay the groundwork for further studies to develop corrective strategies based on either diet modulation or the addition of specific key components during in vitro oocyte maturation to improve oocyte competence during the NBS.

## Figures and Tables

**Figure 1 animals-12-02108-f001:**
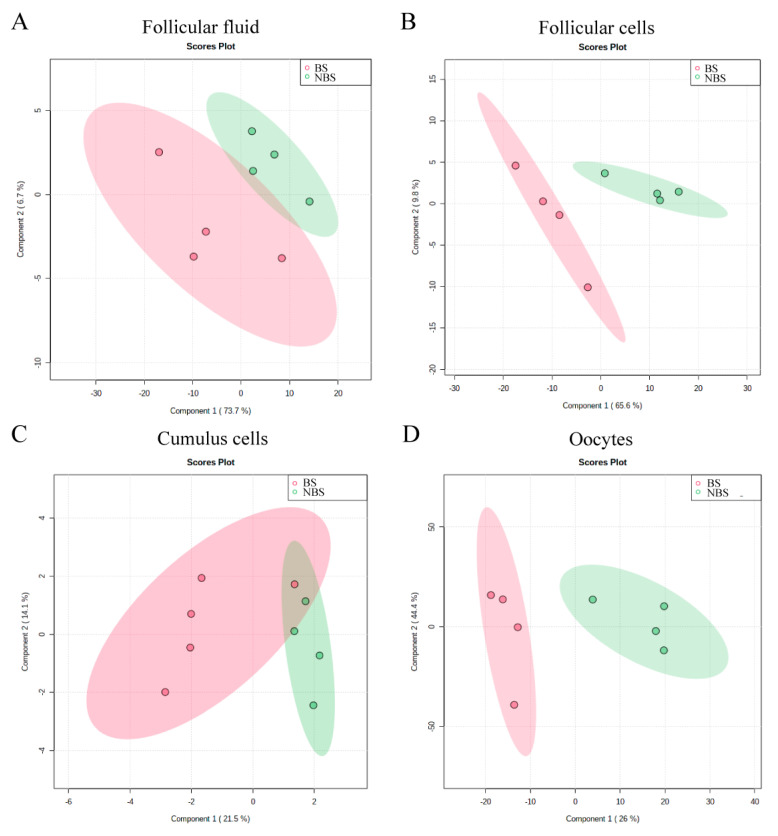
Score plot of samples collected during the breeding season (BS) and the non-breeding season (NBS): (**A**) follicular fluid; (**B**) follicular cells; (**C**) Cumulus cells; and (**D**) Oocytes.

**Figure 2 animals-12-02108-f002:**
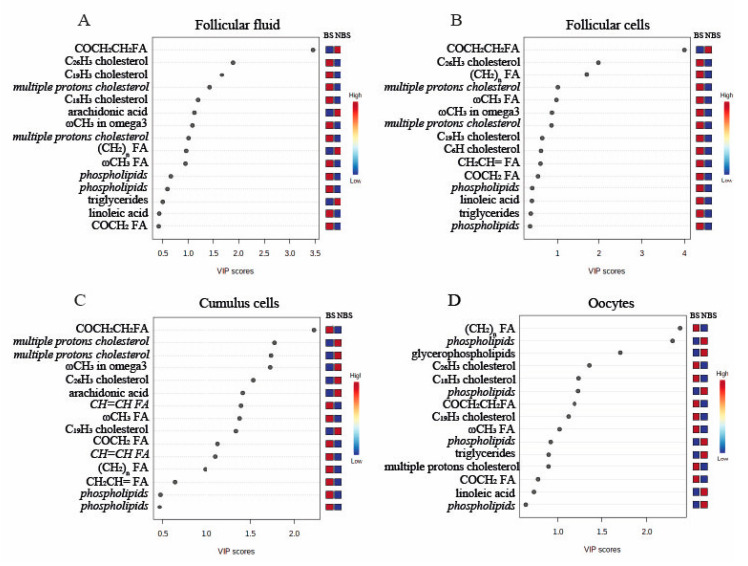
Variable importance in projection (VIP) showing the 15 proton signals corresponding to metabolites with the highest variations between the breeding (BS) and non-breeding (NBS) seasons in: (**A**) follicular fluid; (**B**) follicular cells; (**C**) cumulus cells; and (**D**) oocytes. Metabolites with more than one proton present in the VIP plot are represented in italics.

**Table 1 animals-12-02108-t001:** Differences in apolar metabolite content between seasons in follicular fluid (FF), follicular cells (FC), cumulus Cells (CC), and oocytes (OO). ▲ higher concentration in the NBS vs. BS; ▼ lower concentration in the NBS vs. BS.

	FF	FC	CC	OO
Triglycerides	▲	▼		▲
Cholesterol	▼	▼	▲	▼
Phospholipids	▼	▼	▼	▲
Omega-3 FA	▼	▼	▼	▼
Arachidonic acid	▲		▲	

## Data Availability

The data presented in this study are available on request from the corresponding author.

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
