# Peer review of "Seasonal Variations in the Lipid Profile of the Ovarian Follicle in Italian Mediterranean Buffaloes"

_animals, 2022, doi:10.3390/ani12162108_

Round 1

Reviewer 1 Report

The manuscript presents the data about lipid analysis of different matrices of ovarian follicle in buffalo in the context of reproductive seasonality. The profound analysis provided data, which may explain lower breeding and IVF success in this species in non-breeding season and may indicate the direction of further work to improve reproductive technologies in buffalos. The results are interesting and significantly increase knowledge in the field.  

The manuscript is well structured and generally easy to follow. The introduction part provides sufficient background. The aim is clear and the methodology appropriate. The presentation of the results can be improved (see notes). The data are discussed properly.

I believe that after a minor revision this manuscript will be suitable for publication in the Animals.

Minor remarks:

-        It was not stated if ‘follicular cells’ were granulosa cells, theca cells or both. As they were isolated from aspirated follicular fluid, I guess they were granulosa cells, but this should be clarified.

-        Line 87 and further – the abbreviation NEFAs is used wrongly here (saturated fatty acids (NEFAs). Then NEFA appears without full name further and it is finally explained in Line 257. Some abbreviations (e.g.  BSTFA) are used without expansion. The use of abbreviation should be checked thoroughly in the manuscript and corrected when needed.  

-        Line 101 – ‘cyclic animals’ – shouldn’t be ‘cycling animals’?

-        Line 153 – why cleavage rate was checked on day 5th, not day 1-3, as typically? Some embryos may have degenerated in the meantime and become undistinguishable from degenerated non-fertilized oocytes, lowering the cleavage rate.

-        Figure 1 – quality must be improved, the legend and axis descriptions are totally non-readable. Why there are five red dots on the plot C?

-        Figure 2 – It is written: ‘The metabolites for which more than 1H proton was present in the VIP plot are reported in italics.’ but there is no italics in the plots.

Author Response

Minor remarks:

  • It was not stated if ‘follicular cells’ were granulosa cells, theca cells or both. As they were isolated from aspirated follicular fluid, I guess they were granulosa cells, but this should be clarified.

 Dear reviewer, this is a fair point that you raise. However, in our opinion it is more correct to use the more general term follicular cells, for the cells collected from the follicular fluid. The majority of the cells will certainly be granulosa cells, but we cannot exclude the presence of some theca or cumulus cells. Hopefully this will clarify our choice and preference to describe the cells as follicular cells. 

  • Line 87 and further – the abbreviation NEFAs is used wrongly here (saturated fatty acids (NEFAs). Then NEFA appears without full name further and it is finally explained in Line 257. Some abbreviations (e.g.  BSTFA) are used without expansion. The use of abbreviation should be checked thoroughly in the manuscript and corrected when needed.  

We have checked the manuscript throughout for the correct use of abbreviations and thank the reviewer for this  notification.   

  •        Line 101 – ‘cyclic animals’ – shouldn’t be ‘cycling animals’?

 This is correct, we have changed it accordingly. 

  •        Line 153 – why cleavage rate was checked on day 5th, not day 1-3, as typically? Some embryos may have degenerated in the meantime and become undistinguishable from degenerated non-fertilized oocytes, lowering the cleavage rate.

 In our IVEP buffalo system cleavage rate is assessed on day 5 (day 0 = IVF) concomitant to the change of culture, also to minimize the number of embryo manipulations. We are aware that most laboratories carry out the assessment earlier, but the day 5 score has been our standard procedure since a long time. In addition, an early cleavage assessment may be less reliable in buffalo because some embryos are difficult to pick up (in some cases you can only see a sort of cleaving sept) whereas they are obviously cleaved later on.

  •  Figure 1 – quality must be improved, the legend and axis descriptions are totally non-readable. Why there are five red dots on the plot C

we have improved the quality of the figure as suggested, you will find it in the updated version of the paper. As far as the number of dots is concerned, it simply derives from the fact that an additional sample has been analyzed and it seemed unfair to eliminate it. We believe that this is only anti-aesthetic but does not compromise the result in any way

  •        Figure 2 – It is written: ‘The metabolites for which more than 1H proton was present in the VIP plot are reported in italics.’ but there is no italics in the plots.

the corrected version of the figure has been added. Thanks for noticing

Reviewer 2 Report

I suggest entering the latitude of the study location.

Author Response

We thank the reviewer for this comment, we have added latitude and longitude at line 99.

Reviewer 3 Report

The manuscript “Seasonal variations in the lipid profile of the ovarian follicle in Italian Mediterranean Buffaloes” aimed to evaluate the influence of season on the lipid profile of the ovarian follicle in the Italian Mediterranean Buffalo. The authors presented a clear introduction, identifying the hypothesis of the study and its importance. In the introduction, I only suggest that the authors show more clearly the originality of the work, that is, would it be the type of buffalo? or the quantification of lipids in the reproductive and non-reproductive phases? In the material and methods: How were the types of lipids identified? Explain further in the methodology section. In the results and discussion: Are the blastocyst rates found at present in accordance with the literature? It would be interesting to discuss this information in the discussion section of the manuscript.

Author Response

We thank the reviewer for the thorough revision and will answer the raised points below:

According to the introduction: This manuscript describes for the first time the lipid composition at the different levels of the follicle in buffaloes The originality of the work consists in the evaluation of the lipid profile of the ovarian follicle in buffalo according to season in the attempt to reveal causes of the reduced oocyte competence eroded during the non-breeding season. We believe that this aim has been clearly stated in the introduction (lines 65-68 and 84-86).. 

We have adjusted the description of methods for lipid identification in the material and methods section (Lines 186-200)

Finally, as suggested, a comment has been added regarding the previous literature in relation to the blastocyst rate (Lines 297 - 301)

Reviewer 4 Report

In this paper, Kosior and colleagues analyzed the lipidomic profile in many matrices (follicular fluid, oocytes, cumulus and follicular cells) from fermale Italian buffalo, during both breeding (BS) and non-breeding seasons (NBS). They found many differences, as higher content of triglycerides and lower concentration of cholesterol and phospholipids in the follicular fluid  during the NBS. Thus, the authors concluded that the reduced buffalo oocyte competence during the NBS may be accounted to differences in such lipid composition compared to BS. 

The paper is well written and conducted, however some revisions should be addressed prior its acceptance in Animals:

- The authors should correlate their results to the hormonal profile (not only the hormones of the hypothalamus-hypophysis-gonad axis, but also those regulating energetic balance, as TSH, T3, T4, insulin, and so on)  as, notoriously, it extremely influence reproductive activity;

- Figure 1 is blurred and difficult to read;

- My main concern regards the final aim proposed by the authors, namely the use of strategies to improve buffalo fertility during the NBS: don't the authors think that this may be forcing the natural reproductive activity of these animals? Or, be an additional source of stress for them, that are already enough exploited for economic reasons?

Author Response

  •  The authors should correlate their results to the hormonal profile (not only the hormones of the hypothalamus-hypophysis-gonad axis, but also those regulating energetic balance, as TSH, T3, T4, insulin, and so on)  as, notoriously, it extremely influence reproductive activity;

-Many thanks for your positive comments and thorough review. The originality of the current work was, like mentioned by the reviewer, to evaluate the lipid profile at different levels of the follicle during the breeding and non-breeding season. Unfortunately, We cannot correlate our results with the hormonal profile, because the biological matrices analyzed in this work originated from slaughterhouse ovaries. The collection of extra samples from buffaloes, would require at least one year to be able to include both seasons in the analysis. Therefore, we cannot provide information on the hormonal profile of animals, but we will certainly consider this idea for future studies. 

  • Figure 1 is blurred and difficult to read

As suggested figure 1 was improved

  •  My main concern regards the final aim proposed by the authors, namely the use of strategies to improve buffalo fertility during the NBS: don't the authors think that this may be forcing the natural reproductive activity of these animals? Or, be an additional source of stress for them, that are already enough exploited for economic reasons?

We agree that in order to distribute calvings throughout the year and to meet the market demand buffaloes are forced to conceive during the non-breeding season. However, the out of season mating strategy has been a practice for over 30 years. At present breeding is highly dependent on genetic progress and the utilization of reproductive technologies is fundamental for farm profitability. In the current scenario, the optimization of productive and reproductive performance can lead to an improvement of the conditions of buffaloes reared in the farms, i.e. less animals raised with consequently better welfare and reduced environmental impact. 

Round 2

Reviewer 4 Report

The paper can be accepted in this form.